# Intelligent Machine Learning: Tailor-Making Macromolecules

**DOI:** 10.3390/polym11040579

**Published:** 2019-04-01

**Authors:** Yousef Mohammadi, Mohammad Reza Saeb, Alexander Penlidis, Esmaiel Jabbari, Florian J. Stadler, Philippe Zinck, Krzysztof Matyjaszewski

**Affiliations:** 1Petrochemical Research and Technology Company (NPC-rt), National Petrochemical Company (NPC), P.O. Box 14358-84711, Tehran, Iran; 2Department of Resin and Additives, Institute for Color Science and Technology, P.O. Box 16765-654, Tehran, Iran; mrsaeb2008@gmail.com; 3Department of Chemical Engineering, Institute for Polymer Research (IPR), University of Waterloo, Waterloo, ON N2L 3G1, Canada; 4Biomimetic Materials and Tissue Engineering Laboratory, Department of Chemical Engineering, University of South Carolina Columbia, Columbia, SC 29208, USA; JABBARI@cec.sc.edu; 5College of Materials Science and Engineering, Shenzhen Key Laboratory of Polymer Science and Technology, Guangdong Research Center for Interfacial Engineering of Functional Materials, Nanshan District Key Lab for Biopolymers and Safety Evaluation, Shenzhen University, Shenzhen 518055, China; 6Unity of Catalysis and Solid State Chemistry, University of Lille, CNRS, Bât C7, Cité Scientifique, 59652 Villeneuve d’Ascq Cédex, France; philippe.zinck@univ-lille1.fr; 7Department of Chemistry, Carnegie Mellon University, Pittsburgh, PA 15213, USA

**Keywords:** microstructure, Kinetic Monte Carlo, living copolymerization, olefin block copolymers, artificial intelligence, ethylene, machine learning, genetic algorithms

## Abstract

Nowadays, polymer reaction engineers seek robust and effective tools to synthesize complex macromolecules with well-defined and desirable microstructural and architectural characteristics. Over the past few decades, several promising approaches, such as controlled living (co)polymerization systems and chain-shuttling reactions have been proposed and widely applied to synthesize rather complex macromolecules with controlled monomer sequences. Despite the unique potential of the newly developed techniques, tailor-making the microstructure of macromolecules by suggesting the most appropriate polymerization recipe still remains a very challenging task. In the current work, two versatile and powerful tools capable of effectively addressing the aforementioned questions have been proposed and successfully put into practice. The two tools are established through the amalgamation of the Kinetic Monte Carlo simulation approach and machine learning techniques. The former, an intelligent modeling tool, is able to model and visualize the intricate inter-relationships of polymerization recipes/conditions (as input variables) and microstructural features of the produced macromolecules (as responses). The latter is capable of precisely predicting optimal copolymerization conditions to simultaneously satisfy all predefined microstructural features. The effectiveness of the proposed intelligent modeling and optimization techniques for solving this extremely important ‘inverse’ engineering problem was successfully examined by investigating the possibility of tailor-making the microstructure of Olefin Block Copolymers via chain-shuttling coordination polymerization.

## 1. Introduction

The ubiquity of polymers in daily life, especially in packaging and consumer products, gives them a special importance among different types of materials [1,2,3]. However, the rapid growth of both humanity and the wealth level has led to an ever-increasing demand for resources. Since the mid-1980s, more polymers have been produced in volume than steel (the second-highest-volume material). Among polymeric materials, approximately 70% are thermoplastics, 50% of which are polyolefins; the latter part is dominated by far by polyethylene (PE), polypropylene (PP), and their copolymers. Decades of research have developed important and widely used materials from high-pressure polymerization (low-density polyethylene (LDPE) [4]), (catalyzed) low-pressure polymerization (high-density polyethylene (HDPE), linear low-density polyethylene (LLDPE), and PP [5,6,7,8,9,10,11,12]), and single-site-catalyzed polymerization (metallocene HDPE/LLDPE/PP [13,14]). The property range of polyolefins has been significantly expanded by the more recent and seminal introduction of chain-shuttling polymerization by Dow Chemical Company, where a chain-shuttling agent (CSA) can transport a chain from one catalyst to another, thus allowing for the production of block copolymers inexpensively on an industrial scale [15]. In fact, although the first catalyst is capable of engaging a large number of comonomer units, the second one incorporates small amounts of comonomer. Hence, two distinct types of polyethylene chains, i.e., soft and hard copolymers, are simultaneously formed due to the difference in comonomer consumption tendency of the catalysts. Cross-shuttling of active centers among different living and dormant chains in the reaction medium via CSA molecules results in the synthesis of multi-block polyethylene chains. This invention permits designing olefin block copolymers (OBCs) with properties that allow for using not only different grades as commodity polymers but also as specialty engineering applications [16].

The multi-block structure of OBCs yields a dual character of thermoplastic processability as well as elastomeric solid state properties. More importantly, the properties can be tuned by varying the concentration and type of the catalysts and monomers, as well as the concentration of the chain-shuttling/transfer agent, ultimately leading to different comonomer contents of the hard and soft segments as well as their average lengths [15,17]. These multifaceted properties can be tuned over a wide range, but this fine-tuning is not that trivial, as the first attempt for a usable OBC required 1600 syntheses via high throughput setups with fast molar mass distribution and thermal analyses [15]. The reason behind this is that a whole set of independently controllable factors interact with each other non-trivially and non-intuitively and, consequently, simple predictions of molecular architecture from the synthesis variables are not readily possible. Hence, in comparison to classical polyolefin synthesis, the complexities of OBC polymerization require a highly advanced understanding of polymerization. This was first attempted by Zhang et al. [18], who proposed a kinetic model for chain-shuttling polymerization that was able to correlate average molecular weight, comonomer content, and number of blocks with such synthesis factors as monomer ratio, catalyst composition, and CSA level. While this model was successful in obtaining a rough idea of the structure of the OBC, more powerful tools are required for gaining an in-depth understanding of the microstructure. This has been achieved by recent Kinetic Monte Carlo (KMC) simulations of chain-shuttling polymerizations [19,20,21,22,23,24].

Due to the multifarious types of reactions occurring simultaneously, including cross-propagation and cross-shuttling reactions, setting up these simulations is not an easy task. The KMC simulation was designed to track every monomer, CSA, and catalyst molecule as well as every single chain in the virtual reactor in order to obtain the microstructural features of the OBC. These features are represented by average number of linkage points per chain (*LP*), ethylene sequence length (*ESL*) of hard and soft blocks, average degree of polymerization (*DP_n_*) of hard and soft blocks, longest ethylene sequence length (*LES*) of hard and soft blocks, comonomer content (*C8%*) of hard and soft blocks, and hard block percent (*HB%*) [19,20,21,22,23,24].

While the results on the microstructure of OBCs provided by the KMC simulator are believed to be very accurate, the developed KMC algorithm cannot suggest polymerization variables that can yield a certain desired microstructure. In other words, the KMC simulator receives *X* (*x*_1_, *x*_2_, *x*_3_, …, *x*_n_) as an input of polymerization variables and yields microstructural arrays of *Y* (*y*_1_, *y*_2_, *y*_3_, …, *y*_m_), i.e., the direction is *X→Y* (modeling). For practical applications, however, and in order to address a much more complex and interesting problem, engineers need copolymers with certain microstructures, which cannot be simply deduced from this kind of algorithm or by other means (e.g., deterministic models) due to the complexity of the microstructure. In other words, there is no such thing as a single-valued functional relationship, i.e., a one-to-one relationship between polymerization factors and microstructural features. Hence, the inverse pathway, i.e., selecting synthesis conditions from desired microstructural properties in the direction *Y*→*X* (optimization), constitutes an ill-posed problem.

Classical mathematical modeling methodologies are frequently very complex, time-consuming, and ill-conditioned when confronting very complex systems. The advent of Artificial Intelligence (AI) Modeling and Optimization techniques has opened new possibilities to scientists/engineers working on nonlinear processes. The use of Artificial Intelligence approaches with evolutionary learning in different areas of Science and Engineering, and notably in Materials Science [25,26,27,28,29,30,31], is offering a viable pathway to mimicking the evolution of complex system performance. This makes AI-based methods good candidates for identifying the mechanisms and inter-relations of complex polymerization kinetics. Since polymerization reactions proceed by a series of probability-controlled steps, and at any given time, new molecules are generated, converted, activated, deactivated, and transferred from one class/situation to another, polymerization systems can be better understood and, consequently, their behavior better tracked, by using evolutionary approaches.

The aim of the current study is the establishment and development of unique and versatile modeling and optimization tools capable of handling precise predictions and intricate manipulations of microstructural features of complex macromolecules via amalgamation of KMC simulation approaches and Computational Intelligence techniques. Making this possible requires a deeper understanding of the interplay between different phenomena in the reacting system, culminating in the inter-relationships between polymerization recipes/conditions and final microstructural properties. Two powerful and effective tools, including an Intelligent Modeling Tool (IMT) and an Intelligent Optimization Tool (IOT), are successfully developed and introduced. To construct and implement these tools, the KMC simulation approach has been hybridized with appropriate Artificial-Intelligence-based modeling and optimization techniques. The modeling and optimization of Chain-Shuttling Coordination (co)polymerization (invented by Dow Chemical Company) for the production of OBCs have been selected as a sufficiently complex case study to ‘challenge’ the use and validity of the proposed tools.

## 2. Model Development

The KMC simulator developed previously [19,20,21,22,23,24] acted as a virtual reactor to synthesize chains and look at microstructural aspects of OBCs. The microstructural variables of OBCs can be classified into two categories: (i) those that directly correspond to topological and architectural characteristics of chains, such as *LP*, *DP_n_^SOFT^*, *DP_n_^HARD^*, *ESL^SOFT^*, and *ESL^HARD^*; and (ii) those that indirectly determine the ultimate properties of OBCs, such as *LES^SOFT^*, *LES^HARD^*, *C8%^SOFT^*, *C8%^HARD^*, and *HB%*. There is some evidence that thermal, rheological, mechanical, as well as phase separation properties of OBCs in the melt-state are governed by the second category; for instance, crystallization is controlled to a large extent by *HB%*, *LES^SOFT^*, and *LES^HARD^* of OBCs. Hence, 10 kinetic parameters (two homo-propagation, two cross-propagation, two shuttling to virgin CSA, two shuttling to polymeryl CSA, and two transfer to hydrogen rate constants) as well as the amounts of seven reactants (a solvent, two catalysts, two monomers, hydrogen, and CSA) lead to a set of 10 (output) responses (the abovementioned microstructural features) calculated from six distribution functions (molar mass distribution of both blocks, block number distribution of both blocks, comonomer distribution of both blocks), which are partially dependent on each other. This alone provides sufficient evidence for why simple single-valued one-to-one relationships cannot be found for a complex system such as OBC synthesis.

The hybridization of KMC with machine learning tools enables the prediction and tailoring of the microstructure and ultimate properties of OBCs. In this regard, two different intelligent tools were developed: (i) an Intelligent Modeling Tool (IMT); and (ii) an Intelligent Optimization Tool (IOT). The IMT is a hybrid of the KMC simulator with an intelligent modeler, such as an Artificial Neural Network (ANN) or a Fuzzy Logic System, while the IOT is an intelligent optimizer, combining KMC with, for example, a Genetic Algorithm (GA) or Swarm Intelligence. Figure 1 provides a view of how the functions of IOT and IMT are intertwined.

Application of the IMT and IOT makes it possible to produce new grades of OBCs, with preset characteristics, without numerous trials but only via computer simulations ‘reverse-engineering’ the correct recipe to produce the desired OBC. This kind of approach-finding way to connect 10 microstructural features with highly complex inter-relations of seven synthesis factors, or to determine the possible synthesis variables to obtain a desired set of microstructural features, was tested for the OBC synthesis in references [19,20,21,22,23,24]. The basic scheme developed here can be used in numerous additional ways, spanning from biology to chemistry to physics, provided that some understanding of a possible mechanism exists that can link the input and output variables. These (simulation) calculations can be completed within an acceptable run time (for instance, ca. 45 years in case of KMC versus a few seconds in case of the IMT [22]).

The IMT (*X→**Y*) combines the advantages of ANN and KMC methodologies in a synergistic manner. In a nutshell, the IMT relies on calculating a number of viable scenarios. For instance, polymerization recipes and conditions (*X*) are systematically defined to cover the whole variable space considered. Then, the KMC simulator calculates the microstructural features (*Y*) of all defined scenarios separately, and these are subsequently used to train several ANNs. Obviously, the KMC simulator and intelligent modeler interact in an offline manner. This results in various black boxes, which can intelligently predict any microstructural pattern with high accuracy within the covered factor space. Considering the time-intensiveness of the KMC-algorithm, the IMT offers a fast solution to obtain the required simulation results, which ultimately allows for determining a significantly wider range of polymerization recipes with their corresponding microstructural patterns, while in parallel determining the relationships between operational conditions and microstructural properties.

In the case of the IOT (*Y*→*X*), the KMC simulator is continuously recalled by the GA optimizer via an online route, in obvious contrast to the IMT step (*X*→*Y*), for which the KMC-ANN ‘modeler’ works offline. The GA optimizer randomly generates a number of polymerization recipes and sends the recipes for error evaluation to the KMC simulator. When the KMC simulator “runs” these recipes/conditions, it resembles a virtual reactor that individually receives polymerization recipes as inputs and visualizes microstructural features as outputs. Then, it returns the resulting microstructural characteristics, i.e., *LP*, *DP_n_^SOFT^*, *DP_n_^HARD^*, *ESL^SOFT^*, *ESL^HARD^*, *LES^SOFT^*, *LES^HARD^*, *C8%^SOFT^*, *C8%^HARD^*, and *HB%*, to the GA optimizer online through a closed-loop circuit. The GA optimizer subsequently invokes genetic operators, including selection, sorting, mating, crossover, and mutation, based on the feedback it received to generate the next generation (polymerization recipes); finally, the next generations are considered accordingly by sending them to the KMC simulator and receiving the corresponding microstructural characteristics; these steps are iterative. The process continues until a generation is evolved by the IOT, which contains the recipe required for production of the target OBC having a predefined (desirable) microstructure, i.e., the global optimum. Thus, the IOT allows for ‘dialing-in’ variables and obtaining the required synthesis conditions, which is the inverse way to the above-described IMT approach. The output of the IOT is either a unique recipe or several polymerization recipes (Pareto fronts) corresponding to the preset target microstructural patterns. As a result, well-defined microstructures can be tailored depending on customer demand.

Two separate computer programs were written in the PASCAL programming language (Lazarus 1.2.4 IDE) and compiled into 64-bit executable codes using FPC 2.6.2. The first program, i.e., the intelligent modeler, was established based on Artificial Neural Networks as a powerful black-box modeling technique and appropriately set to recalling our in-house KMC simulator code in an offline mode. On the other hand, the second code, i.e., the intelligent optimizer, was written based on the Non-dominated Sorting Genetic Algorithm (NSGA-II); NSGA-II is an extremely (we would like to think, the most) powerful heuristic multi-objective search strategy to communicate with the KMC simulator via an online route [32,33,34,35]. A subroutine based on the “Mother-of-all Pseudo-Random Number Generators” algorithm was employed to produce the required random numbers during the modeling and optimization steps [36]. The random number generation subroutine satisfied the tests of uniformity and serial correlation with high resolution. The cycle length of the random number generator was 3 × 10^47^. All modeling and optimization steps were performed with a desktop computer with Intel Core i7-3770K (3.50 GHz), 32 GB of memory (2133 MHz), under the Windows 7 Ultimate 64-bit operating system.

## 3. Results and Discussion

The power of the IMT (*X→**Y*) in modeling the microstructure of OBCs has already been recently confirmed by visualizing new grades of OBCs [22]. The “fingerprint” of chain-shuttling copolymerization is envisioned using the developed IMT in terms of architecture-related (*LP*, *DP_n_^SOFT^*, *DP_n_^HARD^*, *ESL^SOFT^*, and *ESL^HARD^*) and property-related (*LES^SOFT^*, *LES^HARD^*, *C8%^SOFT^*, *C8%^HARD^*, and *HB%*) characteristics in a phase diagram like that shown in Figure 2. Figure 2 is like an operational ‘map’ for or ‘window’ into the process. Three-dimensional plots of hard and soft segments for each microstructural feature, for instance *DP_n_^SOFT^* and *DP_n_^HARD^*, are plotted first and subsequently intersected and projected onto a two-dimensional surface. Then, the intersections are determined, at which the studied specific microstructural characteristic takes on the same values for soft and hard blocks, and separately marked by different colors, e.g., the thick blue line denotes the intersection of soft and hard blocks’ lengths (*DP_n_^SOFT^* = *DP_n_^HARD^*). Obviously, the contribution of hard and soft characteristics is inversely changing in the ‘shadowed’ and ‘unshadowed’ areas (see the legends of Figure 2). For instance, the area distinguished by blue color denotes *DP_n_^SOFT^* > *DP_n_^HARD^*, while the uncolored area denotes *DP_n_^SOFT^*<*DP_n_^HARD^*. By putting all these microstructural information pieces together, a master diagram is obtained. Such a master diagram provides a fast survey of the architecture-property topological trends. For a more detailed look at such relationships, i.e., a deeper understanding of changes in the blocky nature of architecture-related and property-related microstructural features of OBCs, and for fine-tuning such characteristics, one can “zoom in” to the three-dimensional plots or corresponding contour plots. For instance, five points, A, B, C, D, and E, in the master plot are randomly considered and their polymerization recipes are read from the master plot. Eventually, the KMC simulator gives microstructural features under the conditions proposed by the IMT, corroborating the authenticity of predictions based on the Artificial Intelligence approach.

Table 1 summarizes the randomly selected points in Figure 2 representing five OBCs and their topology- and property-related characteristics yielded by the IMT and KMC operators. It can be recognized that the IMT has appropriately learned the polymerization behavior and successfully predicted both types of molecular characteristics of randomly selected OBCs, as ‘approved’ by the KMC simulator. A closer look at the statistics in Table 1, bearing in mind the very low errors in predictions of the IMT, assures that such an intelligent machine can reliably be applied in ‘anticipating’ two categories of molecular features, leading to sophisticated architecture-property correlations. The IMT consists of 10 individual prediction mechanisms, each responsible for one molecular characteristic of the OBC, which are integrated into an intelligent toolbox towards the prediction of molecular characteristics of OBCs with ‘surgical’ precision.

The points A–E are several typical examples of OBCs “synthesized”, “characterized”, and identified by the IMT. In the master plot (Figure 2 and Table 1), different classes of OBCs are distinguished, which are yielded from a feed having a monomer molar ratio (MR) of 0.75 and different catalyst compositions (CCs) and CSA levels. These grades are different from each other considering their dissimilar blocky natures. The x-axis of Figure 2 refers to the ratio of the two different catalysts, where a low catalyst composition means having more of the catalyst-producing hard blocks. The log(CSA) level on the y-axis refers to the amount of chain-shuttling agent in the reaction and, thus, to the probability of shuttling events. As can be observed, 10 key characteristics (see Table 1) determining the blocky nature of OBCs are compared in Figure 2. All these together provide an OBC with an “identification card”, which is unique to this special recipe. Exploiting these unique capabilities of the IMT, it is possible to develop new grades of OBC suited to different applications.

Several more results stemming from the master plot (Figure 2) need to be discussed. For instance, while it is counterintuitive that soft blocks (with higher comonomer content) can have a longer ethylene sequence than hard blocks, it becomes immediately obvious that such a possibility exists only for high log(CSA) levels, which in turn leads to very short blocks (few monomer units). This in combination with the higher concentration of soft-block-producing catalyst leads to much longer soft blocks than hard blocks, which means that the average ethylene sequence in the hard block is short, as it cannot be longer than the block itself. Catalyst compositions close to zero lead to much longer hard than soft blocks and vice versa for CC close to 1, as the CSA causes shuttling of the chain to a random catalyst center, which in the case of CC being very different from 1 means that the block lengths are very different from each other. Thus, the possibilities of producing OBCs with previously unexplored structures as well as exact prediction of both their architecture- and property-related molecular features are good examples of the exceptional capabilities of the IMT.

Similar to the case of the IMT, but through a completely different route, the reliability of the IOT has also been tested. For this purpose, two tailored OBCs, OBC1 and OBC2 shown in Table 2, are defined and “animated” using the IOT. OBC1 is a multi-block copolymer in which the weight percent of hard blocks has to take the value of 30; and, at the same time, the average *ESL* of hard blocks is 10 times bigger than the corresponding parameter defined for the soft blocks. In contrast, OBC2 is a highly uniform OBC having a hard block percent of 45. It is worth mentioning that, in a highly uniform OBC, 1-octene units are distributed evenly in both soft and hard blocks. The objectives assigned to each case, as well as the Pareto optimal fronts representing the best solutions with optimal operational conditions, are provided in Table 2. As can be seen, virtual synthesis of OBC1 and OBC2 required concurrent control of three and seven microstructural characteristics of the copolymer chains, respectively, as can be realized by the defined objectives in each case. The IOT iteratively generates and analyzes billions of solutions to find the chain microstructure closest to the target OBC1 and OBC2. To validate the authenticity of the IOT results, the optimum polymerization recipes obtained in each case were entered into the KMC simulator to virtually synthesize OBC1 and OBC2 and determine their actual microstructural characteristics. As can be seen in Table 2, errors in predicting target microstructural features are below 3.0% for almost all quantities. The percentage of error in prediction of microstructural features (highlighted in Turquoise) is defined as the ratio of the difference between the outcomes of the IOT (highlighted in yellow) and KMC (highlighted in green) under the conditions proposed by the IOT (numerator) to the difference between minimum and maximum values of that characteristic for stochastically synthesized macromolecules (denominator) multiplied by 100.

The microstructural characteristics of OBC1 and OBC2 are shown in Figure 3 and Figure 4, respectively. As can be observed, both the evolution and end-of-batch characteristics of the intelligently synthesized OBCs are precisely and thoroughly predicted and depicted.

Based on these findings, it is now possible to precisely model and optimize the synthesis of olefinic block copolymers by using the developed intelligent tools. The IOT and IMT make it possible to determine the features required to synthesize a polymer with the desired molecular structure or to visualize the factor space for an overview of the possible combinations of molecular features as a function of polymerization variables.

The approach introduced herein is obviously bound to the specific catalyst/chain-shuttling system for producing OBCs. However, it is clear that the method works sufficiently well for OBC synthesis, which is one of the most complex syntheses from a reaction kinetics point of view. Considering this, the IMT and IOT can readily be adapted to many other polymerization schemes by modifying the reaction kinetics model in the KMC simulator. Furthermore, any scheme that links any given (input) X to any given (response) Y could be modeled by a modified version of the KMC simulator.

## 4. Conclusions

Two powerful tools were introduced and successfully implemented to model and optimize the microstructural aspects of complex macromolecules. The newly developed computational techniques were established based on hybridization of molecular simulation approaches and Machine Learning techniques. The strategy made it possible to construct intelligent modeling and optimization tools capable of learning and decision-making. Undoubtedly, when applying these tools, polymer reaction engineers not only can effectively discover the complex inter-relationships between polymerization conditions and final architectural characteristics, but will also have the opportunity to adjust rather precisely the polymerization inputs in an attempt to synthesize predefined microstructures in detail. Chain-shuttling coordination copolymerization, an intricate polymerization system, has been chosen as the first test case to challenge the proposed intelligent modeling and optimization tools. The results obtained clearly showed that the IMT was capable of meticulously patterning the molecular landscape of OBCs in terms of operating conditions, including monomer molar ratio, catalyst composition, and CSA level. IMT was effectively put into practice to ‘crack’ the inter-relationship between operating conditions and micro-molecular characteristics and/or final properties of interest. By superimposing all microstructural information pieces together, a master diagram is obtained that provides a fast survey of the recipe-architecture-property topological trends. In contrast, the IOT was able to accurately predict the input/operating factors in response to predefined micro-molecular/architectural characteristics of the target OBC chains. To precisely evaluate the accuracy and performance of the proposed IOT, two target OBCs were designed first. Then, the IOT was implemented and applied to concurrently optimize three and seven molecular characteristics of the predefined OBC chains, respectively. The results obtained demonstrated that the IOT was able to successfully handle multi-objective optimization problems and simultaneously control various micro-molecular/architectural characteristics, resulting in negligible errors calculated for the objective functions.

Although the unique capabilities of the proposed techniques were successfully tested and examined with a complex living shuttling coordination copolymerization case study, they can be employed by both academic and industrial experts to model and optimize all types of macromolecular reactions and other reactive systems. Both the IMT and IOT, as intelligent computational tools, have the potential to guide polymer chemists and engineers towards the realization of advanced ‘living and thinking’ materials.

## Figures and Tables

**Figure 1 polymers-11-00579-f001:**
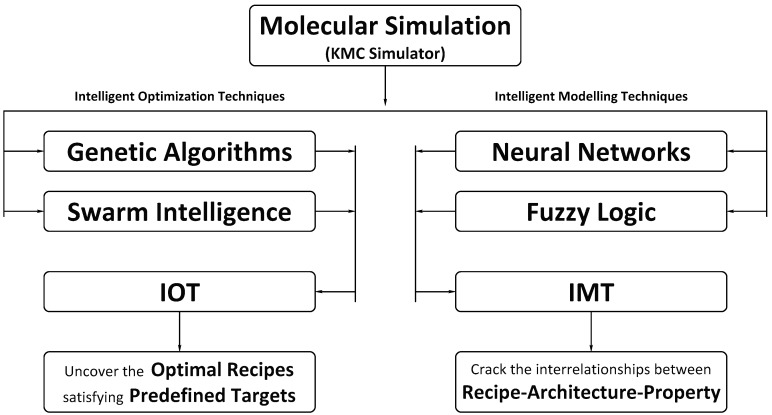
Illustrative description of the Intelligent Modeling Tool (IMT) and the Intelligent Optimization Tool (IOT) developed for tailoring the microstructure of olefin block copolymers (OBCs). KMC, Kinetic Monte Carlo.

**Figure 2 polymers-11-00579-f002:**
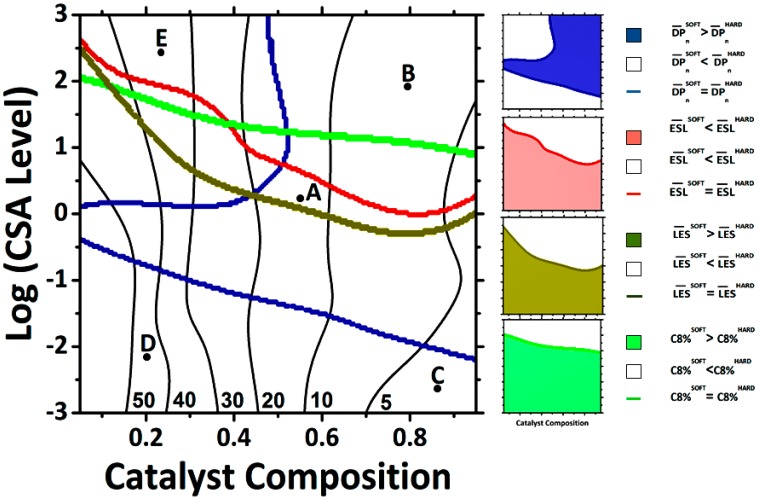
A “fingerprint” of chain-shuttling copolymerization decoded by the IMT (which visualizes different classes of OBCs all having a monomer molar ratio of 0.75, but a different blocky nature). Values (50 to 5) appearing at the bottom of the plot close to the black curves represent copolymers having a specified hard block percent (*HB%*). CSA, chain-shuttling agent; DP, average degree of polymerization; ESL, ethylene sequence length; LES, longest ethylene sequence length.

**Figure 3 polymers-11-00579-f003:**
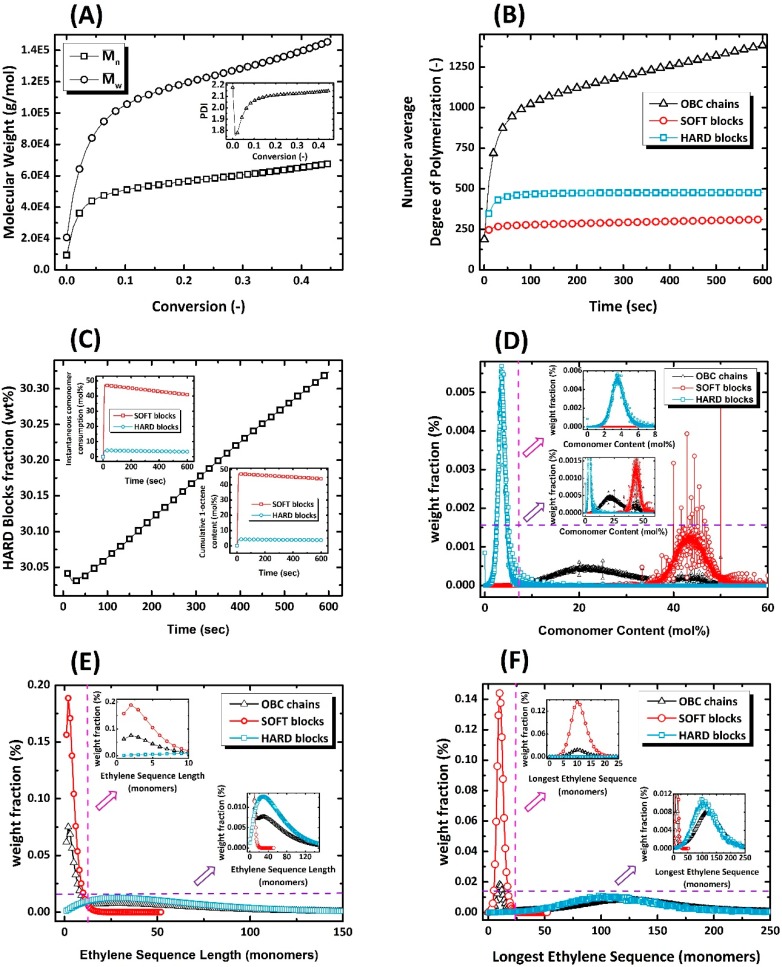
The identification card of intelligently synthesized OBC1 demonstrating all instantaneous and cumulative microstructural features of virtually produced chains.

**Figure 4 polymers-11-00579-f004:**
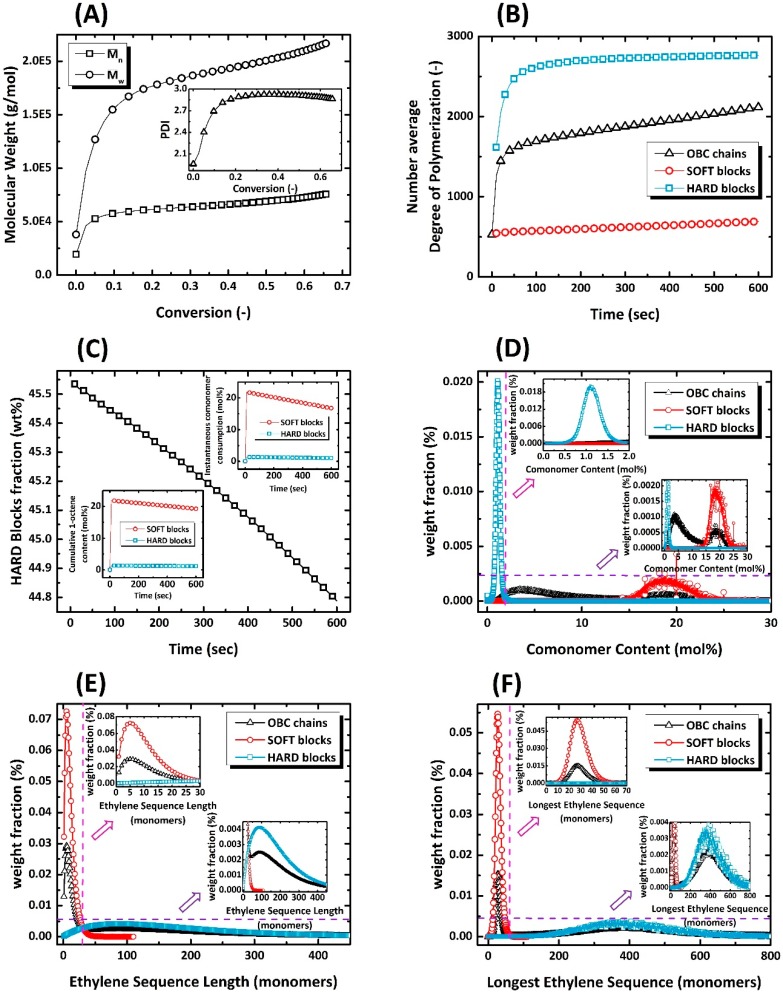
The identification card of intelligently synthesized OBC2 demonstrating all instantaneous and cumulative microstructural features of virtually produced chains.

**Table 1 polymers-11-00579-t001:** Typical OBCs randomly selected in the ‘master plot’ of the IMT and synthesized by the KMC simulator under specified operating conditions together with their topology- and property-related molecular characteristics.

	Topology-Microstructural Characteristics					Property-Microstructural Characteristics				
	*DP_n_^SOFT^*	*DP_n_^HARD^*	*ESL^SOFT^*	*ESL^HARD^*	*LP*	*C8%^SOFT^*	*C8%^HARD^*	*LES^SOFT^*	*LES^HARD^*	*HB%*
**Point A: {MR = 0.75 and CC = 0.539 and log(CSA) = 0.313}**										
**KMC**	172.88	30.65	25.10	28.92	36.51	3.55	0.70	96.46	55.19	13.30
**ANN**	169.31	122.07	24.67	31.76	37.26	3.12	0.51	87.49	61.51	14.17
**Error %**	0.01	2.45	0.92	0.34	1.01	0.88	1.07	2.61	0.14	1.53
**Point B: {MR = 0.75 and CC = 0.792 and log(CSA) = 1.886}**										
**KMC**	122.41	92.11	23.63	10.52	64.87	2.17	9.23	72.12	16.45	5.97
**ANN**	160.86	102.50	24.58	13.87	63.72	2.62	8.87	78.00	18.84	6.63
**Error %**	0.10	0.28	2.04	0.41	1.55	0.92	2.02	1.71	0.05	1.16
**Point C: {MR = 0.75 and CC = 0.859 and log(CSA) = −2.594}**										
**KMC**	3301.32	13068.35	46.70	649.53	0.01	2.94	0.17	306.40	3183.62	2.52
**ANN**	3555.20	13176.48	47.54	666.29	0.64	3.10	0.25	313.26	3249.55	4.53
**Error %**	0.67	2.90	1.80	2.03	0.85	0.32	0.45	1.99	1.47	3.52
**Point D: {MR = 0.75 and CC = 0.202 and log(CSA) = −2.184}**										
**KMC**	1873.32	18132.94	20.32	434.77	0.14	5.03	0.26	111.81	1842.66	44.80
**ANN**	1967.33	18094.88	20.84	456.82	0.33	5.26	0.19	125.13	1927.41	45.15
**Error %**	0.25	1.02	1.12	2.68	0.26	0.47	0.39	3.87	1.89	0.61
**Point E: {MR = 0.75 and CC = 0.229 and log(CSA) = 2.397}**										
**KMC**	94.37	148.96	23.26	12.54	71.58	6.24	13.02	34.72	9.61	35.05
**ANN**	67.66	117.61	22.84	16.01	72.82	6.00	12.53	30.50	17.83	35.52
**Error %**	0.07	0.84	0.90	0.42	1.67	0.49	2.75	1.23	0.18	0.82
**LV**	5.13	7.12	2.34	2.19	0.00	1.89	0.12	3.29	5.84	1.35
**HV**	38085.60	3736.87	48.97	825.43	74.28	50.72	17.90	347.30	4480.15	58.38

LV and HV are the lowest and highest values of a given response among simulated scenarios, respectively. MR and CC stand for Monomer Molar Ration and Catalyst Composition, respectively. ANN, artificial neural network.

**Table 2 polymers-11-00579-t002:** The chain microstructure of the OBC1 and OBC2 macromolecules tailored using the IOT. The constraints/objectives used in the multi-objective optimization, Pareto fronts showing the best solution, and optimum input variables proposed by the IOT for the synthesis of OBC1 and OBC2 are provided. Microstructural features controlled by the IOT highlighted in yellow; KMC optimizer outputs obtained at identical optimum polymerization recipes highlighted in green; errors in the IOT predictions highlighted in Turquoise.

**(A) OBC1**				**(B) OBC2**			
**Objective 1:**		MIN|HB%−30|		**Objective 1:**	$MIN\left|\left(\frac{{\bar{DP}}_{n}^{Soft}}{1+{\bar{N}}_{C_{8}}^{Soft}}\right)-ESL^{Soft}\right|$		
**Objective 2:**		$MIN\left|\left(\frac{ESL^{Soft}}{ESL^{Hard}}\right)-0.1\right|$		**Objective 2:**	$MIN\left|\left(\frac{{\bar{DP}}_{n}^{Hard}}{1+{\bar{N}}_{C_{8}}^{Hard}}\right)-ESL^{Hard}\right|$		
				*N_C8_*: the average number of comonomer units per block.			
				**Objective 3:**	MIN|HB%−45|		
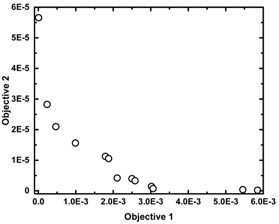				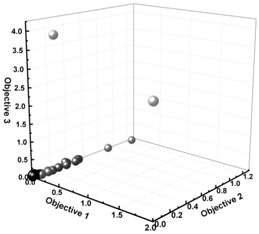			
**Optimum Input Variables**				**Optimum Input Variables**			
MR		0.213890		MR		0.44058	
CC		0.44755		CC		0.29194	
Log(CSA Level)		−0.79549		Log(CSA Level)		−1.41288	
**Optimum Responses**				**Optimum Responses**			
	**IOT**	**KMC**	**Error (%)**		**IOT**	**KMC**	**Error (%)**
***C8*%*^SOFT^***	46.9435	44.0155	5.99667	***C8*%*^SOFT^***	16.1221	19.2357	6.37665
***C8*%*^HARD^***	3.66274	3.73534	0.40827	***C8*%*^HARD^***	1.28743	1.14977	0.77419
***HB%***	30.0000	30.3236	0.56760	***HB%***	44.9994	44.7886	0.36974
***ESL^SOFT^***	2.78577	2.53169	0.54488	***ESL^SOFT^***	5.33127	5.58988	0.5546
***ESL^HARD^***	27.8735	29.5826	0.20761	***ESL^HARD^***	85.9224	89.3081	0.41127
***DP_n_^SOFT^***	327.719	310.251	0.04587	***DP_n_^SOFT^***	678.959	692.488	0.03553
***DP_n_^HARD^***	362.897	475.781	3.02657	***DP_n_^HARD^***	2853.94	2765.08	2.38255
***N_C8_^SOFT^***	123.617	136.605	3.46589	***N_C8_^SOFT^***	126.314	133.236	1.84699
***N_C8_^HARD^***	20.5723	17.7899	0.17786	***N_C8_^HARD^***	32.2173	31.8509	0.02342

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
