# Peer review of "Intelligent Machine Learning: Tailor-Making Macromolecules"

_polymers, 2019, doi:10.3390/polym11040579_

Round 1
Reviewer 1 Report
In this manuscript, Kris Matyjaszewski, Mohammadi, and their coworkers demonstrate a novel machine learning (ML) method for the discovery of new polymer materials. Machine learning represents a very promising methodology to significantly alleviate the efforts of material scientists to make new materials with predictable properties. However, few studies were reported on applying ML to polymerizations such as control radical polymerizations and chain shutting reactions. The study in this manuscript serves as a bridge between ML and optimizing polymerization conditions. Specifically, the authors harnessed both kinetic Monte Carlo simulation and Artificial Intelligence techniques to elucidate not only the connections between polymerization conditions and microstructure, but also to predict the best copolymerization conditions for desired microstructure. As a proof of concept, chain shuttling polymerization was used to evaluate the method established by the authors. In general, the design of the simulation is exceptional. The data is solid and conclusive. Writing is smooth to follow. Therefore, I trust this manuscript will attract wide interest from not only polymer chemists but also theoretic chemists. I recommend acceptance of this article to Polymers after the authors address my comments below.
Comments
While the study is very interesting, the references are quite old. For the first sentence of introduction which illustrates the ubiquity of polymers, a relevant reference should be cited. Highly suggest adding Sumerlin et al. Prog. Polym. Sci., 2019, 89, 61-75. doi.org/10.1016/j.progpolymsci.2018.09.006.
Did authors try to simulate controlled radical polymerization (CRP), especially those with ATRP and RAFT mechanism? It will be very exciting to see the application of their ML method into those CRP systems because control radical polymerization especially ATRP has now become an essential tool for almost every polymer chemist.
Author Response
The authors would like to extend their appreciation to respected Reviewer for his/her meticulous reading and sharing invaluable comments. We tried our best to precisely study and appropriately address all concerns one-by-one.
Modifications per Reviewer #1 can be seen in Yellow in the revised version of the manuscript. Following you can find a point-by-point reply to all comments of Reviewer #1.
Point 1:
While the study is very interesting, the references are quite old. For the first sentence of introduction which illustrates the ubiquity of polymers, a relevant reference should be cited. Highly suggest adding Sumerlin et al. Prog. Polym. Sci., 2019, 89, 61-75. doi.org/10.1016/j.progpolymsci.2018.09.006.
Response 1:
Several references were added as recommended by the reviewer.
Point 2:
Did authors try to simulate controlled radical polymerization (CRP), especially those with ATRP and RAFT mechanism? It will be very exciting to see the application of their ML method into those CRP systems because control radical polymerization especially ATRP has now become an essential tool for almost every polymer chemist.
Response 2:
We have already started applying and implementing both IMT and IOT to simulate, model, and optimize CRP systems. Undoubtedly, amalgamation of molecular simulation methods and ML techniques can be of significant importance in case of RAFT, ATRP, and NMP systems and is capable of revealing more insights about CRP systems and cracking the complexities of the interrelationships between polymerization parameters and final micro-molecular characteristics/material properties.
Definitely, it is our pleasure to have the great opportunity to collaborate with Reviewer #1 in simulation, modeling, and optimization of CRP systems applying molecular simulation approaches and ML techniques.
Reviewer 2 Report
The authors performed simulations regarding the copolymerization of soft and hard blocks. The manuscript, in its current form, cannot be accepted for publication. There are several parts, which should be improved significantly. In detail, these are:
Generally, the manuscript is too long and not focussed. The introduction is too long providing information, which are not required. It should be more focussed on the important parts.
The results and discussion part is also too long and, thus, the main information of the manuscript are not clear to me as reader. Please focus on the main idea and I suggest to add a supporting information document with the not important information.
An outlook section is missing, which would be nice.
The conclusion part is very general. Thus, the reader get not the main idea by reading this section. The conclusion part should contain the main idea without reading the rest of the manuscript.
The abstract is also too general.
What is not clear to me: What is the new information provided by the manuscript? In particular, if one would compare it to the previous work of the same authors (references 16-20; in particular 17). There are very similar figures, the same monomers etc. etc. And by reading the manuscript, I did not get the main new results. Please clearify that and compare your new results with the already exisiting ones. Please indicate, what is new. This goes in hand with a more focussed manuscript, since the main idea is not clear to me.
The simulation are nice, but a comparison with experimental data would be very good and would improve the manuscript significantly. Since highly experienced synthetic groups contributed to the paper, a comparison between simulation and experiments should be possible.
Author Response
The authors would like to extend their appreciation to reviewer #2 for his/her meticulous reading of the ms and for making very good suggestions. We tried our best to address all comments, as described in detail below.
Modifications per Reviewer #2 can be seen in Turquoise in the revised version of the manuscript. Following you can find a point-by-point reply to all comments of Reviewer #2.
Point 1:
The conclusion part is very general. Thus, the reader get not the main idea by reading this section. The conclusion part should contain the main idea without reading the rest of the manuscript.
Response 1:
Thanks for your helpful comment. Several lines of more information were added in the conclusion section.
Point 2:
What is not clear to me: What is the new information provided by the manuscript? In particular, if one would compare it to the previous work of the same authors (references 16-20; in particular 17). There are very similar figures, the same monomers etc. etc. And by reading the manuscript, I did not get the main new results. Please clarify that and compare your new results with the already existing ones. Please indicate, what is new. This goes in hand with a more focused manuscript, since the main idea is not clear to me.
Response 2:
Thanks for your comment. In our previous papers, we have mostly focused on molecular simulation of chain shuttling coordination polymerization systems, i.e. the development of an appropriate KMC simulator for virtual synthesis of OBCs. Recently, we have started to implement artificial intelligence techniques in macromolecular reaction engineering. The fact is that there exist several well-developed intelligent modeling and optimization techniques/algorithms which can be effectively applied in case of polymerization systems. In our previous studies, we have separately applied evolutionary modeling methods and heuristic optimization techniques to handle some typical macromolecular reactions/processes.
The main purpose of the current study, however, is to build a general perspective in terms of the importance and position of machine learning techniques in both modeling and optimization of complex macromolecular reaction engineering. In fact, we have attempted to illustrate the outstanding capabilities of intelligent modelers and intelligent optimizers in handling a typical intricate polymerization reaction engineering case study. Hence, not only the readers can experience the exact implementation mechanism of IMT and IOT in case of macromolecular reactions but also they will be able to distinguish the differences and outstanding aspects of each method. In addition, we have amalgamated molecular simulators with both intelligent modelers and optimizers, which makes it possible to compare the capabilities of the proposed tools. The cases have been studied in this work are completely different from the ones already investigated in our previous papers. All in all, the current work provides a complete and concise description of the development/implementation of the intelligent machine learning based modeling and optimization techniques in macromolecular reactions.
Point 3:
The simulation is nice, but a comparison with experimental data would be very good and would improve the manuscript significantly. Since highly experienced synthetic groups contributed to the paper, a comparison between simulation and experiments should be possible.
Response 3:
As it has been already mentioned, the KMC simulator has been mainly established based on (1) the experimental data reported by Arriola et al. and (2) the kinetic model proposed by Zhang et al. The members of both teams are expert chemists and/or engineers from the same company, i.e. The DOW Chemical Company. As you know, OBC is invented by this company and the details are highly confidential. So, the simulator has been established based on the experimental data reported by the mentioned teams. The simulator has been capable of precisely predicting the reported experimental data (the development of the simulator has been already explained in detail in our previous works). Also, the developed intelligent modeling and optimization tools (IMT and IOT) based on machine learning approaches have been developed based on widely accepted standard algorithms.
Reviewer 3 Report
The paper entitled « Intelligent machine learning : tailor-making macromolecules » by Mohammadi et al. proposes the use of theoretical modeling and optimization techniques in order to obtain precise predictions regarding the microstructural features of complex macromolecules, such as the olefin block copolymers (OBC).
The paper is well written, clear and the conclusions are supported by the results. Moreover, this study is very interesting not only for the polymer chemists working at the lab scale but also for industrial engineers.
I have only minor suggestions:
- the phrase between the lines 75-79 is quite complicated.
- in table 1, “LV” and “HV” stands for what?!
- I don’t know if the colors from table 2 are accepted by the editor (maybe the authors can color only the numbers)
Author Response
The authors would like to extend their appreciation to respected Reviewer for his/her meticulous reading and sharing invaluable comments. We tried our best to precisely study and appropriately address all concerns one-by-one.
Modifications per Reviewer #3 can be seen in Bright Green in the revised version of the manuscript. Following you can find a point-by-point reply to all comments of Reviewer #3.
Point 1:
The phrase between the lines 75-79 is quite complicated.
Response 1:
For the purpose of clarification, it was paraphrased.
Point 2:
In table 1, “LV” and “HV” stands for what?!
Response 2:
LV and HV stand for the lowest and highest values of a given response among simulated scenarios, respectively. It has been explained in the revised version of the manuscript.
Point 3:
I don’t know if the colors from table 2 are accepted by the editor (maybe the authors can color only the numbers).
Response 3:
Thanks for your helpful suggestion. Definitely, we will check it with the editor.
Round 2
Reviewer 2 Report
I am still not convinced that the manuscript can be accepted for publication. The basic problem is the reliability of the experiments. The data, which were utilized for the design of the AI, are only partially accessible. The programm behind cannot be reviewed by me since the programm is also not accessible.
Thus, the data presented can be right; however, it is not possible for me to accept the paper since I am not able to see any data behind.
If the details are highly confidential as claimed by the authors, they should think about if publication is the right way from them.